Growth of 48 built environment bacterial isolates on board the International Space Station (ISS)

Coil David A. 1 coil.david@gmail.com
Neches Russell Y. 1
Lang Jenna M. 1
Brown Wendy E. 1 2
Severance Mark 2 3
Cavalier Darlene 2 3
Eisen Jonathan A. 1 4 5 jaeisen@ucdavis.edu
1 Genome Center, University of California , Davis, CA , United States
2 Science Cheerleader , Philadelphia, PA , United States
3 SciStarter.com , Philadelphia, PA , United States
4 Department of Medical Microbiology and Immunology, University of California , Davis, CA , United States
5 Department of Evolution and Ecology, University of California , Davis, CA , United States
Baltrus David
Electronic publication date: 2016 Mar 22
Publication date: 2016
Volume: 4
Electronic Location ID: e1842
Received 2016 Jan 16; Accepted 2016 Mar 2
Copyright: ©2016 Coil et al.
Copyright year: 2016
Copyright holder: Coil et al.
License: This is an open access article distributed under the terms of the Creative Commons Attribution License, which permits unrestricted use, distribution, reproduction and adaptation in any medium and for any purpose provided that it is properly attributed. For attribution, the original author(s), title, publication source (PeerJ) and either DOI or URL of the article must be cited.
License URL: https://creativecommons.org/licenses/by/4.0/

Keywords: Bacterial growth, International space station, Built environment, Microgravity, Space, Non-pathogenic

Funding: Alfred P. Sloan Foundation International Space Station Research Competition (ISSRC) This work was funded by both the Alfred P. Sloan Foundation, through their “Microbiology of the Built Environment” program as well as through Space Florida via their International Space Station Research Competition (ISSRC). The funders had no role in study design, data collection and analysis, decision to publish, or preparation of the manuscript.

==============================
Background. While significant attention has been paid to the potential risk of pathogenic microbes aboard crewed spacecraft, the non-pathogenic microbes in these habitats have received less consideration. Preliminary work has demonstrated that the interior of the International Space Station (ISS) has a microbial community resembling those of built environments on Earth. Here we report the results of sending 48 bacterial strains, collected from built environments on Earth, for a growth experiment on the ISS. This project was a component of Project MERCCURI (Microbial Ecology Research Combining Citizen and University Researchers on ISS).

Results. Of the 48 strains sent to the ISS, 45 of them showed similar growth in space and on Earth using a relative growth measurement adapted for microgravity. The vast majority of species tested in this experiment have also been found in culture-independent surveys of the ISS. Only one bacterial strain showed significantly different growth in space. Bacillus safensis JPL-MERTA-8-2 grew 60% better in space than on Earth.

Conclusions. The majority of bacteria tested were not affected by conditions aboard the ISS in this experiment (e.g., microgravity, cosmic radiation). Further work on Bacillus safensis could lead to interesting insights on why this strain grew so much better in space.

Introduction

From 2012 to 2014, we conducted a nationwide citizen science project, Project MERCCURI http://spacemicrobes.org/, aimed at raising public awareness of microbiology and research on board the International Space Station (ISS). Project MERCCURI (Microbial Ecology Research Combining Citizen and University Researchers on the ISS) was a collaborative effort involving the “microbiology of the Built Environment network” (microBEnet) group, Science Cheerleader, NanoRacks, Space Florida, and SciStarter. One of the goals of Project MERCCURI was to examine how a number of non-pathogenic bacteria associated with the built environment would grow on board the ISS compared to on Earth.

Most previous work growing bacteria in space has focused on species known to contain pathogenic strains (e.g., Escherichia coli (Klaus et al., 1997; Brown, Klaus & Todd, 2002) and Pseudomonas aeruginosa (Crabbé et al., 2011; Kim et al., 2013a), and much less attention has been paid to the non-pathogenic microbes that surround us. An understandable bias towards pathogens and pathogenic pathways is highlighted by work on topics such as biofilm formation (Kim et al., 2013b; McLean et al., 2001), antibiotic resistance/production (Benoit et al., 2006; Juergensmeyer, Juergensmeyer & Guikema, 1999; Lam et al., 2002 reviewed in Klaus & Howard, 2006), and virulence (Nickerson et al., 2000; Hammond et al., 2013).

Although concern about pathogens in spacecraft is certainly warranted, it should be emphasized that the ability of a pathogen to survive outside a host and the ability to infect a host are both, at least in part, dependent on the existing community of non-pathogenic microbes in those locations. For example, the infectivity of some pathogens has been shown to be very dependent on the host microbiome (e.g., Schuijt et al., 2015; Ichinohe et al., 2011; Van Rensburg et al., 2015; Reeves et al., 2011). Therefore, it is important to understand the entire microbial ecosystem of spacecraft. Indeed, in recent years, several culture-independent studies have examined the microbiome of the ISS (Castro et al., 2004; Venkateswaran et al., 2014; Moissl et al., 2007), including another part of Project MERCCURI (J Lang et al., 2015, unpublished data). These studies have shown, not surprisingly, that the microbiome of the ISS bears a strong resemblance to the microbiome of human-associated built environments on Earth. Therefore, it is of interest to see how microbes from human-associated environments behave in space.

For this study, samples from human-associated surfaces (e.g., toilets, doorknobs, railings, floors, etc.) were collected at a variety of locations around the United States, usually in collaboration with the public. A wide variety of bacteria were cultured from these samples, and 48 non-pathogenic strains were selected for a growth assay comparing growth in microgravity on the ISS and on Earth.

Materials and Methods

Sample collection

Samples were collected from built environment surfaces throughout the United States on cotton swabs (Puritan 25-806 2PC) and mailed (usually overnight) to the University of California Davis where they were transferred to lysogeny broth (LB) plates. Colonies were chosen for further examination based on maximizing morphological variation. Each chosen colony was double-dilution streaked (two rounds of streak plates) and then the identity determined by direct PCR and Sanger sequencing using the 27F (5′-AGAGTTTGATCMTGGCTCAG-3′) and 1391R (5′-GACGGGCGGTGTGTRCA-3′) primers (see Dunitz et al., 2015 for details). Sanger sequences were trimmed and aligned using Geneious (Kearse et al., 2012). The resulting consensus sequence was identified through a combination of BLAST (Altschul et al., 1990) and building phylogenetic trees using the Ribosomal Database Project (RDP) website (Cole et al., 2014). The 48 candidates for spaceflight were chosen on the basis of biosafety level (BSL-1 only), taxonomic variety, and human interest. In the absence of international standards, the biosafety level of each organism was determined by searching the American Biological Safety Association (ABSA) risk group database, the American Tissue Culture Collection (ATCC), the Deutsche Sammlung von Mikroorganismen und Zellkulturen (DSMZ), and other public databases. An organism was removed from consideration if it was listed as BSL-2 or higher in any country or collection in the world. Human interest was an arbitrary set of criteria such as unusual physiology, catchy name, or a memorable original isolation source.

Growth experiment

A set of bacterial plates were created for each aspect of the study: growth in microgravity on the ISS (space plates), or growth on Earth (ground plates). The plates were created using clear agar to facilitate optical density (OD) measurements. 1.5 g of Gelzan CM agar (Sigma-Aldrich, St. Louis, MO, USA) was added to 1 liter of lysogeny broth (LB). Each well of a flat-bottomed 96-well plate (Costar, Corning, NY, USA) was plated with 200 µl of agar. The plates were flamed to remove bubbles and incubated for 48–72 h at room temperature (∼20 °C) to ensure sterility before adding bacteria. Fresh overnights of each bacterial isolate were diluted to .01 OD600 and made into 8% glycerol stocks. For plating, 10 µl of each thawed stock dilution was added to two wells per 96 well plate. Six replicate plates were made. The bacteria were placed into different locations on each plate in order to account for drying at the edges or any other positional effects on the plates. The plates were then sealed with adhesive polypropylene film (VWR #60941-072), into which a grid of micron-diameter holes were cut with a laser to allow for airflow. The ground plates were stored at −80 °C at UC Davis, and the space plates were mailed on dry ice to the National Aeronautics and Space Administration (NASA) Johnson Space Center in Houston, TX before transfer (at −80 °C) to Cape Canaveral, FL for launch.

This payload was flown on the CRS-3 launch of the Space Exploration Technologies (SpaceX) Dragon spacecraft, on a Falcon 9 v1.1 rocket which successfully launched April 18, 2014. After six days, the space plates were removed from the MELFI (Minus Eighty Lab Freezer For ISS) and partially thawed. However, technical problems arose and the space plates were placed back into the MELFI until December 8, 2014. At that time, all three plates were thawed and the OD600 of each well (3×3 grid) was measured at time 0 (60 min after removal from the freezer) and then every 24 h for four days. Measurements were performed in a Molecular Devices SpectraMax M5e plate reader which was modified for integration onto the ISS. On these same days, equivalent measurements of the ground plates were taken in a Molecular Devices SpectraMax M5e plate reader at UC Davis. The exception to this was the initial partial thawing, which was not replicated with the ground plates since the amount of thaw was not reported by the astronauts. After the experiment, the ground plates were placed back at −80 °C and the space plates were placed back into the MELFI. In February 2015, the space plates were transferred to a −95 °C freezer on board a Dragon spacecraft. The vehicle splashed down in the Pacific Ocean on Feb 10, 2015. The space plates were then mailed to UC Davis on dry ice and were transferred to −80 °C when received.

Once the plates arrived, we thawed all six plates and performed a high-density measurement in a Tecan M200 plate reader. OD600 readings were taken in a 5×5 grid covering the entire well, these 25 measurements were then averaged within each well.

Analysis

For each sample, the averages of the six space replicates and six ground replicates were compared using a Student’s t-test. To correct for multiple hypothesis testing, the p-values were adjusted using the False Discovery Rate (FDR) method (Benjamini & Hochberg, 1995). All raw data, analyses and scripts can be found at https://zenodo.org/record/44661.

Confirmation

In order to confirm that the observed results were not due to contamination of the wells, each of the 12 replicates (six space, six ground) for the three bacteria showing statistically different growth between the ISS and Earth were cultured after the experiment. Bacteria were struck from the wells onto LB-agar plates, then single colonies were grown into overnight cultures. DNA was extracted using a Wizard Genomic DNA Purification kit (Promega, Madison, WI, USA) from each of the 36 cultures (3 bacteria ×12 replicates) and the identity was confirmed by BLAST of the Sanger sequenced PCR product using the 27F and 1391R primers as described above.

Comparison to ISS swab data

The bacterial community on the ISS was recently surveyed by PCR amplification and sequencing of 16S rRNA genes from swabs (Lang et al., unpublished data). We compared the 16S sequence of each of our bacterial isolates to the “representative sequence” from each operational taxonomic unit (OTU) generated from the survey data. A BLASTN search was performed locally and a match was considered to be present in the data when there was 97% identity over at least 250 bp of the rRNA sequence (the amplified fragment is 253 bp).

Results and Discussion

Growth experiments are typically undertaken in liquid media, in part because measuring the optical density of a liquid culture is straightforward. However, liquid cultures present a number of problems in microgravity. Most organisms that passed our screening did not grow well under anaerobic conditions, and thus required some sort of gas exchange with the surrounding air. On the ground, aerobic conditions are easily created by incubating in open or loosely capped vessels. This is impractical and unsafe in microgravity; there is no “safe” orientation in which the liquid will remain in place. We explored several unsuccessful approaches to this problem. For example, we found that gas-permeable plate seals leak when inverted, and their adhesion failed completely after freezing. We also fabricated custom plates with seals made from hydrophobic polydimethylsiloxane (PDMS) with micron-diameter vent holes, but these also leaked when inverted.

We eventually concluded that the design requirements were mutually exclusive; either we could achieve containment for liquid cultures at the expense of aerobic conditions, or we could achieve aerobic conditions at the expense of liquid culture containment. We chose the latter, so our plates were prepared with solid media. Solid media is not traditionally used for OD measurements, and so our results need to be interpreted differently from OD in liquid culture. Using clear agar to maximize transparency, we programmed the plate reader to take OD measurements at nine different locations in each well, each of which was measured twenty five times per observation. The plates were inoculated in a manner intended to create many small colonies (see ‘Materials and Methods’). As these colonies grow, their edges intersect with reading points, and the OD for that point increases in a stepwise fashion. As the colony thickens, the OD gradually increases. OD in liquid media is thought to correspond to scattering of light by individual cells, whereas our measurements correspond to the number, diameter, and thickness of the colonies. The intervals elapsed between occultations of the reading points decrease exponentially, and so the average OD across each well behaves very similarly to traditional observations of log-phase growth in liquid media. However, in the absence of correlation with the gold standard of dilution plate counts, this should be considered as a relative measure of growth. This was validated by repeated growth experiments on Earth, showing normal growth kinetics of colonies grown with this method, a sample dataset is shown in Fig. S1. To our knowledge this is the first use of solid media to measure bacterial growth kinetics in this manner. The data from the different plate readers (Tecan and Molecular Dynamics) was compared at 96 h by plotting the OD600 values against each other. While the concordance was not perfect, there was a strong relationship between the two machines which provided validation of the data from both Molecular Dynamics machines (ground and space).

By this measure, the vast majority of the bacteria (45/48) behaved very similarly in space and on Earth (Table 1). Only three bacteria showed a significant difference in the two conditions; Bacillus safensis, Bacillus methylotrophicus, and Microbacterium oleivorans. As part of double checking these results, we performed Sanger sequencing of rRNA genes from the wells corresponding to each of these species of the space plates and ground plates. A few wells produced mixed Sanger sequence, suggesting the presence of more than one organism in the well. In addition, a couple of wells gave a clear identification of a contaminating organism. We therefore inferred that there had been some contamination of the B. methylotrophicus and M. oleivorans wells. Since the remaining 45 organisms were not tested for contamination, it is possible that some of those represent false negatives. The B. safensis wells were all clear of any signs of contamination.

Table 1 Final growth (OD600) of all 48 strains as measured on Earth at the end of the experiment with a high-density measurement in a Tecan platereader.

Values represent the mean of 6 wells, ± the standard deviation. Difference between space and Earth were determined using a Student’s t-test and the p-values were adjusted for multiple hypothesis testing by using the False Discovery Rate (FDR).

Organism	Location	Source	Mean OD (space)	Mean OD (ground)	FDR p-value	
Bacillus safensis	JPL-NASA (CA)	Mars Exploration Rover	1.44 ± 0.09	0.86 ± 0.07	0	
Bacillus methylotrophicus	Yuri’s Night New York (NY)	Doorknob	1.54 ± 0.05	1.69 ± 0.09	0	
Microbacterium oleivorans	St. Joseph’s Prep (PA)	School mascot	1.61 ± 0.3	1.93 ± 0.14	0.04	
Bacillus atrophaeus	JPL-NASA (CA)	Mars Exploration Rover	1.69 ± 0.05	1.57 ± 0.14	0.07	
Porphyrobacter mercurialis	Pop Warner: Coronado (CA)	Stadium seat	0.85 ± 0.13	1.03 ± 0.2	0.07	
Bacillus flexus	NFL: Tennesse Titans (TN)	Stadium field	1.47 ± 0.24	1.72 ± 0.1	0.07	
Bacillus atrophaeus	Denver Museum of Nature and Science (CO)	Antique microscope	1.62 ± 0.06	1.3 ± 0.25	0.14	
Bacillus altitudinis	Deerfield Academy (MA)	School field	1.23 ± 0.09	1.22 ± 0.15	0.14	
Macrococcus brunensis	WHYY Radio (PA)	Keyboard	1.06 ± 0.15	1.29 ± 0.11	0.14	
Bacillus tequilensis	Today Show (NY)	Candy jar	1 ± 0.21	1.09 ± 0.1	0.14	
Bacillus amyloliquefaciens	NFL: New England Patriots (MA)	Stadium seat	1.41 ± 0.13	1.53 ± 0.12	0.14	
Bacillus subtilis	JPL-NASA (CA)	Robotic arm (Insight)	1.32 ± 0.16	1.08 ± 0.25	0.16	
Micrococcus luteus	NBA: Sacramento Kings (CA)	Sweat mop	1.01 ± 0.08	0.87 ± 0.16	0.21	
Leucobacter chironomi	Davis (CA)	Toilet	1.03 ± 0.27	1.03 ± 0.12	0.21	
Kocuria kristinae	NBA: San Antonio Spurs (TX)	Court floor	1.93 ± 0.06	1.85 ± 0.16	0.21	
Kocuria rhizophila	Yuri’s Night Los Angeles (CA)	Camera	2.01 ± 0.14	1.97 ± 0.19	0.21	
Bacillus stratosphericus	Academy of Natural Science (PA)	Water dish	1.34 ± 0.14	1.1 ± 0.13	0.21	
Bacillus tequilensis	MBA: Philadelphia Phillies (PA)	Dugout	1.41 ± 0.18	1.03 ± 0.15	0.21	
Micrococcus luteus	Pop Warner: Lake Brantley (FL)	Football goalpost	1.71 ± 0.03	1.69 ± 0.06	0.21	
Paenibacillus mucilaginosus	Field Museum (IL)	“Sue” the T. rex	1.57 ± 0.13	1.54 ± 0.14	0.21	
Exiguobacterium sibiricum	AT&T Park (CA)	Second base	1.3 ± 0.23	1.38 ± 0.14	0.21	
Exiguobacterium indicum	NFL: Team from Washington D.C.	Stadium field	1.26 ± 0.16	1.17 ± 0.23	0.21	
Curtobacterium pusillum	UC Davis (CA)	Stadium gate	1.28 ± 0.3	1.49 ± 0.14	0.21	
Kocuria marina	Yuri’s Night North Carolina (NC)	Water Fountain	1.77 ± 0.1	1.73 ± 0.08	0.26	
Bacillus megaterium	The Liberty Bell (PA)	The Liberty Bell	1.38 ± 0.24	1.46 ± 0.15	0.34	
Bacillus lichenformis	NBA: Philadelphia 76ers (PA)	Practice court	1.18 ± 0.13	1.07 ± 0.14	0.34	
Bacillus megaterium	JPL-NASA (CA)	Mars Curiosity Rover	1.6 ± 0.14	1.55 ± 0.16	0.38	
Bacillus subtilus	NBA: Orlando Magic (FL)	Game ball	1.35 ± 0.08	1.17 ± 0.19	0.38	
Arthrobacter nitroguajacolicus	Chapman Hill Elementary (OR)	Stadium field	1.67 ± 0.12	1.76 ± 0.25	0.44	
Bacillus aryabhattai	NFL: Oakland Raiders (CA)	Practice field	1.62 ± 0.3	1.64 ± 0.13	0.44	
Microbacteria arborescens	JPL-NASA (CA)	Viking Mars Orbiter	1.69 ± 0.26	1.59 ± 0.47	0.49	
Bacillus pumilus	JPL-NASA (CA)	Mars Exploration Rover	0.97 ± 0.25	1.26 ± 0.25	0.49	
Paenibacillus elgii	JPL-NASA (CA)	Mars Exploration Rover	1.39 ± 0.3	0.84 ± 0.14	0.49	
Kocuria rosea	JPL-NASA (CA)	Mars Exploration Rover	1.61 ± 0.26	1.53 ± 0.18	0.49	
Bacillus aryabhattai	Pop Warner: Broncos (FL)	Stadium field	1.65 ± 0.28	1.54 ± 0.05	0.49	
Micrococcus yunnanensis	Discover Magazine (WI)	Dictionary	1.68 ± 0.41	1.75 ± 0.23	0.49	
Bacillus amyloliquefaciens	Franklin Institute (PA)	Statue	1.4 ± 0.09	1.38 ± 0.14	0.6	
Bacillus megaterium	Chemical Heritage Foundation (PA)	Antique pressure vessel	1.57 ± 0.43	1.56 ± 0.14	0.61	
Exiguobacterium acetylicum	NFL: San Franciso 49ers (CA)	Stadium field	1.57 ± 0.18	1.53 ± 0.21	0.61	
Bacillus horikoshii	Parkway Middle School (FL)	Banister	1.53 ± 0.34	1.67 ± 0.09	0.61	
Macrococcus equipercicus	Catholic Montessori School (OH)	Floor	0.99 ± 0.19	0.94 ± 0.2	0.64	
Streptomyces kanamyceticus	KARE11 Morning News (MN)	Set kitchen	1.11 ± 0.2	0.92 ± 0.16	0.66	
Pantoea eucrina	Smithsonian Air and Space Museum (D.C.)	Mercury Orbiter	1.57 ± 0.31	1.57 ± 0.09	0.76	
Bacillus horikoshii	Pop Warner: Saints (NJ)	Stadium field	1.64 ± 0.2	1.58 ± 0.07	0.79	
Curtobacterium herbarum	Georgia Tech (GA)	Stadium seat	1.42 ± 0.19	1.5 ± 0.13	0.79	
Bacillus pumilus	Pop Warner: Chittanoga (NY)	Porta-Potty handle	1.17 ± 0.31	1.35 ± 0.12	0.82	
Micrococcus luteus	Pop Warner: Apopka (FL)	Practice mat	0.99 ± 0.27	0.86 ± 0.34	0.82	
Bacillus marisflavi	Pop Warner: PeeWee Bengals (NC)	Stadium field	1.66 ± 0.19	1.61 ± 0.26	0.82	

This Bacillus safensis strain was collected at the Jet Propulsion Laboratory (JPL-NASA) on a Mars Exploration Rover before launch in 2004. As part of standard Planetary Protection protocols, all surface-bound spacecraft are sampled during the assembly process and those strains are then saved for further analysis. We obtained this strain as part of a collection of JPL-NASA strains to send to the ISS (Table 1). In this experiment, Bacillus safensis grew to a final density of ∼60% higher in space than on the ground, with very little variation between replicates (Fig. 1). The genome sequence of this strain, Bacillus safensis JPL-MERTA-8-2 has just been published (Coil, Benardini & Eisen, 2015) and may contain clues as to why this strain behaved so differently in space.

Figure 1 Growth (OD600) over time of Bacillus safensis JPL-MERTA-8-2 in space (green) and on Earth (brown).

Values represent the mean of six wells, ± the standard deviation.

It is perhaps no surprise that most built environment-associated bacteria behave very similarly on the ISS as on Earth. After all, the ISS is a home and office of sorts, with environmental conditions very similar to a building on Earth with the exception of gravity. The ISS is maintained at around 22 °C with a relative humidity of around 60% and pressure and oxygen concentrations very close to those at sea level on Earth. Additionally, this experiment did not provide enough time to study the long-term adaptation of bacteria to the environment on board the ISS.

A related project from our lab has examined the microbial community already present on the ISS (Lang et al., unpublished data). Given that the ISS appears to harbor similar microbes to built environments on Earth, we also asked if there were close relatives to our 48 bacteria already present on the ISS. The vast majority (39/48) of our bacterial species were found in the existing microbial community data which is not surprising given the built environment origins of the isolates. This suggests that our data showing these species growing with similar kinetics on space and on Earth is potentially relevant to the biology of the microbial communities already present on the ISS.

Supplemental Information

Figure S1 Example growth kinetics of solid-agar growth assay

Values represent the mean of 25 measurements, on each of 25 spots, in each of 8 wells. These bacterial strains were grown for 96 h with OD600 measurements taken every 15 min in a Tecan F200 platereader.

Click here for additional data file.

The authors would like to thank the many people who contributed to Project MERCCURI and to the processing of these cultures including Pop Warner, Science Cheerleaders, Nanoracks LLC, Space Florida, SciStarter, Jennifer Flanagan, Ruth Lee, Hannah Holland-Moritz, Alex Alexiev, Madison Dunitz, and Holly Bik. In particular, thanks to Carl Carruthers at Nanoracks LLC who shepherded the payload through all challenges. This article was written using the Authorea scientific writing platform.

Additional Information and Declarations

Competing Interests

Author Contributions

Data Availability

Jonathan Eisen is an Academic Editor for PeerJ. Darlene Cavalier is the founder of Scistarter and Science Cheerleader. Wendy E. Brown and Mark Severance are also affiliated with these organizations.

David A. Coil conceived and designed the experiments, performed the experiments, analyzed the data, wrote the paper, prepared figures and/or tables, reviewed drafts of the paper.

Russell Y. Neches conceived and designed the experiments, performed the experiments, analyzed the data, prepared figures and/or tables, reviewed drafts of the paper.

Jenna M. Lang conceived and designed the experiments, performed the experiments, analyzed the data, reviewed drafts of the paper.

Wendy E. Brown, Mark Severance, Darlene Cavalier and Jonathan A. Eisen conceived and designed the experiments, reviewed drafts of the paper.

The following information was supplied regarding data availability:

Zenodo: https://zenodo.org/record/44661#.VpfzVFKGODA and https://zenodo.org/record/46934.

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
