# Peer review of "Growth of 48 built environment bacterial isolates on board the International Space Station (ISS)"

_PeerJ, doi:10.7717/peerj.1842_

## Round 0.1 · original submission · Minor Revisions

· Academic Editor

Minor Revisions

This manuscript was reviewed by three different individuals, and all were quite enthusiastic about the scope, style, and data. For the most part the reviews reflect cosmetic changes that I agree with, however, I am in agreement with reviewer 3 that the agar growth curve assay should be validated/documented with regard to cell number. Perhaps the focal Bacillus strain plus one other would serve as good subjects for this validation.

Reviewer 1 ·

Basic reporting

No comments

Experimental design

Is this the first time solid media has been used to record growth in 96 well plates? If so, you may want to include a statement in the abstract or introduction.

Validity of the findings

No comment

Comments for the author

This paper was well written and concise. I only have three minor comments.
1.Line 8: The first time I read Science Cheerleaders I thought it was another way of saying "citizen scientist." To make it clear that it is an organization and to keep it consistent with the annotation for Scistarter, I would give the website in the author list, as was done for sciestarter.org.
2. Line 113-114: This statement was clunky. To clarify, I would suggest ""For plating, 10ul of each thawed stock dilution was added to 2 wells per 96 well plate. 6 replicate plates were made." The next sentence nicely explains the randomization of the placement of the bacteria.
3. Line 124: The space plates were partially thawed and re-frozen. Were the plates on earth treated the same way? A partial thaw and re-freeze may affect the growth of some strains and not others. Please clarify.
4. Table 1 legend: Is this the final OD600, or some representation of growth over time? If it is the final OD600, I suggest indicating that this is the terminal reading made when the space plates were back on earth, and not the final OD600 from the space station vs the final OD600 of the readings of the plates grown on earth.

Reviewer 2 ·

Basic reporting

Line 45 (Abstract): “bacterial strain that avoided contamination” is awkward. It might read more clearly to simply say “Only one bacterial strain showed significantly different growth in space.”

Thank you for the well-organized raw data repository. The README file is very clear and easy to follow. Two separate points for the raw data repository:
(1) In the Space vs Ground charts, is the color key the same as Figure 1 (Red = Earth, Green = Space)? I didn’t see this in the README file.
(2) Is the comparison data for the SpectraMax and Tecan plate readers included in the raw data repository? This would be helpful for demonstrating how well the plate readers agreed.

A couple of copy edits:
Line 64 needs ending ) after Pseudomonas citations.
Line 174: programed should be programmed

Experimental design

The methods sections is well-written and supplies a good amount of detail. This information is very helpful for understanding the experimental design, including the particular challenges of the agar-based growth assay and how you addressed them.

Line 91: Glad to see "lysogeny broth". I called it Luria-Bertani broth along with almost everybody else until I learned of its origins in Bertani’s 2004 article.

Line 93: “double-dilution streaked”. I haven’t heard this exact term before. Does this mean you performed two rounds of streak plates, or one streak plate with three phases (the original streak and then two more for the double dilution)?

Validity of the findings

Paragraph starting at line 188: Based on the methods, I understand that only the three strains showing significant differences in growth were screened for contamination in wells. For two of these three, wells showed evidence of contamination, which suggests that perhaps wells for the other 45 strains may have contamination as well. Is it possible that growth differences could be masked by contamination? In other words, perhaps strain A in the “ground” plate grows to a higher density than the same strain in the “space” plate, but the space plate was contaminated with strain B, which is unaffected by microgravity and therefore grows to the same density as strain A?

Lines 188-189 and Table 1: are these data from the 96 hr time point using the Molecular Devices SpectraMax plate reader only? Please include this information about time point and plate reader in the Table legend.

There is a lot of data in the repository for the multiple time points, but it appears that only the final time point is presented in the text and table. Were there any significant differences between “space” and “ground” wells for a particular strain across time points?

Comments for the author

The authors investigate how microbes isolated from human-associated built environments on earth behave on the International Space Station using an agar-based growth assay developed to address the challenges of microgravity. This work emphasizes non-pathogenic strains in order to understand the microbial ecosystem of the ISS built environment.

I enjoyed reading and reviewing this paper. What might seem like a “simple” growth assay is made quite complicated by the microgravity environment of the ISS, and the authors clearly presented the challenges and their solutions.

·

Basic reporting

Minor comments regarding presentation:

Line 51: '...this bacteria… is improper grammar. It should be modified to 'this bacterium' or 'this strain' or 'these bacteria'.

Line 64: 'Pseudomonas aeruginosa' should be used instead of just 'Pseudomonas', particularly given that the authors are referring to species not genera.

Line 65: The authors should replace “normal” with non-pathogenic. There is nothing abnormal about pathogenic microbes, and the fact that the authors bracketed normal with quotation marks, implies they also do not think non-pathogens are more normal than pathogens.

Line 69: The use of 'While' implies a time-dependent comparison, which the authors are not making in this sentence. 'Although' would be more appropriate in this circumstance.

Lines 72-74: This sentence is a bit of a stretch. The referenced article by Kembel et al., 2012 (reference 14 in this manuscript) did not discuss the idea of lack of competition among microbes as being a contributor to the increased pathogen load in ventilated hospitals. Kemble et al. did discuss the idea of reduced dilution of the indoor microbial community by mechanical ventilation compared to window ventilation, but did not invoke any idea of significant interaction among the microbes as being the cause of increased pathogenic species between the two systems. This sentence, as it currently reads is fairly misleading as to the work that was performed and the conclusions drawn in reference 14. The authors should rewrite (or better, omit) this sentence as it does not really support their premise that the microbial community-at-large can affect the persistence and virulence of a pathogen. The premise itself is fine, and supported in the next sentence.

Lines 80-81: 'Therefore' should be followed by a comma, as it is an introductory phrase.

Line 93: What does 'double-dilution streaked' mean?

Line 99: What were the criteria for 'human interest'?

Materials and Methods section: There appeared to be no mention of using uninoculated wells as controls (blanks). The authors stress that they used clear agar, however, LB itself is not optically clear, and would need to be taken into account for the OD measurements to be meaningful. It would be good for the authors to note the inclusion of this control.

The Table 1 legend should indicate that the reported ODs are for the 96 hour time point.

Experimental design

The authors use a non-standard method for measuring OD (for understandable reasons) but do not present any supporting data validating their approach. The authors describe in detail how measurements are taken (lines 174-183) and that a given OD measurement results from a combination of 'number, diameter, and thickness of colonies'. Given the integration of these three variables, what is the relationship between OD and cell number? Traditional (broth culture) OD measurements are compared to dilution counts to establish the relationship between OD and cell number for a given strain, what is the relationship between OD and cell number for a culture growing on a solid medium? In my opinion, this approach could be validated by comparing the OD measurements to dilution plate counts obtained by resuspending the plate grown cells into buffer using a bath sonicator. Clearly this approach is infeasible on ISS, but could be performed on earth to establish the connection between plate OD and cell number. Inclusion of such data would greatly support their conclusions.

Validity of the findings

The authors claim the ISS environment is much like built environments on earth except for the gravitational force and cosmic radiation. They note that temperature and relative humidity are similar. What about pressure and oxygen concentrations? Clearly ISS is maintained within parameters that will support humans, but is the ISS similar to pressure and oxygen concentrations that would be experienced at sea level or in the Rocky Mountains? Because the authors are only comparing growth in Davis, CA to the ISS, the assumption seems to be that B. safensis JPL-MERTA-8-2 will exhibit nearly identical growth dynamics everywhere on earth, as long as the temperature and RH is controlled. Is this true?

---

## Round 0.2 · accepted · Accept

· Academic Editor

Accept

Please be sure to keep on top of the manuscript through publication, to make sure that the reference garbling problem gets fixed.